# Safety and Efficacy of Rivaroxaban as Extended-Phase Anticoagulation in Patients with Cancer and Venous Thromboembolism: A Preliminary Data Analysis from the Mac Project

**DOI:** 10.3390/life12111725

**Published:** 2022-10-28

**Authors:** Enrico Bernardi, Giuseppe Camporese, Cristiano Bortoluzzi, Franco Noventa, Davide Ceccato, Chiara Tonello, Stefania Vohong, Elena Campello, Chiara Simion, Egidio Imbalzano, Pierpaolo Di Micco, Elena Callegari, Paolo Simioni

**Affiliations:** 1Emergency Department, Cà Foncello Hospital, Azienda Ulss 2 Marca Trevigiana, Piazzale Ospedale 1, 31100 Treviso, Italy; 2UO Internal Medicine, Azienda Ospedaliera di Padova, Via Giustiniani 2, 35128 Padova, Italy; 3Internal Medicine, San Giovanni e Paolo Hospital, Azienda Ulss 3 Serenissima, Via Don Tosatto 147, 30174 Mestre, Italy; 4QUOVADIS No-Profit Association, Studio Coppola, Galleria Ezzelino 5, 35139 Padova, Italy; 5Unit of Angiology, Azienda Ospedaliera di Padova, Via Giustiniani 2, 35128 Padova, Italy; 6UO Thromboembolic Disease, Azienda Ospedaliera di Padova, Via Giustiniani 2, 35128 Padova, Italy; 7Department of Clinical and Experimental Medicine, Polyclinic Hospital University of Messina, Via Consolare Valeria 1, 98124 Messina, Italy; 8Internal Medicine, “A. Rizzoli” Polyclinic Hospital, Azienda NA2 Nord, Via Fundera 2, 80076 Lacco Ameno, Italy; 9UO Internal Medicine 2, Cà Foncello Hospital, Azienda Ulss 2 Marca Trevigiana, Piazzale Ospedale 1, 31100 Treviso, Italy

**Keywords:** cancer-related venous thromboembolism, factor Xa inhibitors, extended-phase anticoagulation, real-life

## Abstract

Extended-phase anticoagulation with direct oral Xa inhibitors (OAXI) is suggested in patients with cancer-associated venous thromboembolism (CAT). We report on patients enrolled in the MAC (Monitoring AntiCoagulants) Project, given rivaroxaban as extended-phase anticoagulation after CAT. The primary efficacy outcome was the incidence of symptomatic recurrent VTE; the primary safety outcomes were incidence of major and non-major clinically relevant bleeding, adverse events, and all-cause mortality. The mean patients’ follow-up was 19 months (SD 16); 64/604 (11%) had CAT. Recurrent VTE occurred in 9.3% and in 8.1% of patients with and without CAT (OR 1.2, 95% CI 0.5 to 2.9; *p* = 0.6). Major bleeding occurred in 4.7% and in 2.6%, respectively (OR = 1.8, 95% CI 0.5 to 6.6, *p* = 0.4), and non-major clinically-relevant bleeding in 4.7% and in 4.1% (OR = 1.2, 95% CI 0.3 to 3.9, *p* = 0.7). The relative figures for fatal haemorrhage and all-cause death were 1.6% versus 0%, and 1.6% versus 0.4%. Rivaroxaban appears to be effective and safe as extended-phase anticoagulation in patients with CAT. The mean treatment period was 3-times the standard 6-month course.

## 1. Introduction

Venous thromboembolism (VTE) is a major determinant of morbidity and mortality in cancer patients [1]. Indeed, despite well-conducted anticoagulation with vitamin-K antagonists (VKA), cancer patients with VTE are more likely to develop both recurrent thromboembolic complications and major bleeding events during long-term treatment than those free of malignancy; the event-rate being up to 5-fold higher in patients with advanced cancer [1].

Some 20 years ago, a landmark study demonstrated the efficacy of dalteparin for the treatment of cancer-associated VTE (CAT) [2]. Thereafter, low-molecular-weight heparins (LMWH) replaced VKA in that setting [2,3,4]. Of note, LMWH were also found to be associated with reduced mortality in patients with CAT [5,6]. A main issue with LMWH is long-term compliance with daily subcutaneous injections, both burdensome and inconvenient for the patients [7,8].

In recent years, direct oral Xa inhibitors (OXAI) have been tested in patients with CAT, showing similar efficacy and safety to LMWH [9,10,11,12]. Specifically, according to two meta-analyses of some 3000 patients with CAT, a 6-month treatment with OXAI yields an absolute 3% risk reduction of recurrent VTE, along with an absolute 1% risk increase of major bleeding events, as compared with LMWH [13,14].

Besides, especially from a patient-centred perspective, OXAI appear to be advantageous over LMWH. Indeed, OXAI almost double treatment persistence as compared with LMWH, being cheaper, not needing weight-adjusted dosing, and sparing both injection-related pain and fear [8]. Furthermore, OXAI do not carry the risk of heparin-induced thrombocytopenia.

International guidelines suggest extended-phase anticoagulation with either OXAI or LMWH in patients with CAT who completed the initial 6-month treatment course; however, due to the scarce evidence available, there is very low certainty about this statement [15]. Thus, further information on the outcome of these patients beyond the initial 6-month anticoagulation period would be desirable. 

With this preliminary report we aimed to provide data on patients with CAT, who were given rivaroxaban for extended-phase anticoagulation, over an intended follow-up period of 5 years. To size the proportion of adverse events in CAT patients, we used the correspondent data of patients without CAT, which were taken as a “reference”.

## 2. Methods

The MAC (Monitoring AntiCoagulants) Project is an ongoing prospective-cohort, multi-centre, observational study, held in Italy. The aim of the Project is to collect real-life clinical information in unselected patients given oral anticoagulants for VTE, over a 5-year follow-up period. The study protocol, approved by the Institutional Review Board of each participating centre, is described in detail elsewhere [16].

Briefly, subjects of both sexes, aged 18 years or older, with objectively diagnosed VTE, irrespective of the index event, of the intended treatment duration, and of the type of anticoagulant treatment, are eligible for inclusion. There are no exclusion criteria, except for life expectancy < 6 months, and refusal to sign the informed consent form or to attend the planned follow-up visits. All patients are followed-up prospectively with visits scheduled at 3, 6, and 12 months after the index event, and then annually for up to 5 years.

The primary efficacy outcome is the incidence of symptomatic recurrent VTE; the primary safety outcomes are incidence of major and non-major clinically relevant bleeding, adverse events, and all-cause mortality. The occurrence of any outcomes is assessed at each follow-up visit, or in case of patient-reported suspicion.

All data is recorded in the ad-hoc Electronic Data Collection platform built by the REDCap online facility of QUOVADIS Association. 

## 3. Statistical Analysis

Main demographic and clinical characteristics of patients with and without cancer at baseline were compared by the t-test (continuous variables), or the chi-square test/Fisher’s Exact test (categorical variables). A *p* value < 0.05 was regarded as statistically significant.

Odds ratios (OR) along with their 95% confidence intervals (CI) were estimated by the exact conditional maximum likelihood method. 

Poisson distribution and test-based methods were used to construct the confidence intervals of incidence rates and of the rate ratios.

Continuous variables were described as means plus standard deviations, along with their 25% and 75% quartiles.

Cumulative incidences of VTE and major haemorrhage in patients with or without cancer at baseline were estimated by the Kaplan-Meier method and tested by the log-rank test; patients who died or were lost to follow-up were censored at the time of their last examination.

Outcome analysis was performed on all subjects enrolled in the study. In the case of multiple outcomes occurring to one patient, only the first event was counted for the analysis. No pre-specified subgroup analysis was planned. All calculations were performed with IBM-SPSS version 26.0 (IBM Corp., Armonk, NY, USA).

## 4. Results

Of 604 patients included in this preliminary report, 64 (10.6%) had active cancer at inclusion. The commonest cancer-types encountered were breast (23.6%), multiple myeloma (7.3%), lung (5.5%), kidney (5.5%), and colon-rectum (4.0%). The mean observation period was 18.8 months (SD 16.1; Q1 6, Q3 29 months) for patients with CAT, and 19.8 months (SD 16.2; Q1 6, Q3 29 months) for patients without CAT. The median observation period for patients with and without CAT was 12.5 and 14 months, respectively. Main demographics and metrics are shown in Table 1. Appendix A shows the different cancer types in patients with and without recurrence. Appendix A details the index VTE event in patients with cancer and recurrence.

All patients were treated with rivaroxaban, either 20 mg (92.7%) or 10 mg (6.3%) daily.

### 4.1. Primary Efficacy Outcome

At the time of this analysis, 6 (9.3%) of 64 patients with cancer and 44 (8.1%) of 540 without cancer had experienced at least one episode of recurrent VTE (odds ratio [OR] 1.2, 95% confidence interval [CI] 0.5 to 2.9; *p* = 0.6). The relative time-to-event course is displayed in Figure 1. 

The corresponding incidence rates in patients with and without CAT were 6.0 (95% CI 2.2 to 13.0), and 4.9 (95% CI 3.6 to 6.6) per 100 patient years (Risk ratio [RR], 1.2; 95% CI 0.4 to 2.8; *p* = 0.6); over a cumulative follow-up of 100.3 and 890.8 patient years, respectively.

Pulmonary embolism occurred in 3.1% in patients with malignancy and 1.1% in those without (*p* = 0.2); the respective figures for proximal and distal deep-vein thrombosis being 1.6% vs. 2.0% (*p* = 0.8), and 1.6 vs. 2.4%, (*p* = 0.7).

At the 6-months time point, the standard anticoagulation course for most patients, 5.5% (0.0 to 11.6%) of patients with CAT versus 1.8% (0.6 to 3.0%) without CAT had experienced a recurrent event.

Interestingly, 5 (83.3%) of 6 patients with cancer and recurrence were on rivaroxaban at the time of the event, as compared with 9 (20.5%) of 44 patients without cancer (OR 5.0, 95% CI 1.6 to 15.4; *p* = 0.003). The relative time-to-event course is displayed in Figure 2.

The corresponding incidence rates were 6.6 (95% CI 2.1 to 15.4) and 1.4 (95% CI 0.6 to 2.79) per 100 patient years (RR 4.6; 95% CI 1.2 to 15.4; *p* = 0.002); over a cumulative follow-up of 75.9 and 634.1 patient years, respectively. Of note, all 5 patients with CAT and 5 of those without CAT were taking rivaroxaban 20 mg OD at the time of recurrence, while the remaining 4 were taking 10 mg OD.

### 4.2. Primary Safety Outcome

At the time of this preliminary analysis, 3 (4.7%) of 64 patients with cancer, as compared with 14 (2.6%) of 540 without cancer, had experienced a major haemorrhagic event (OR 1.8, 95% CI 0.5 to 6.6, *p* = 0.4). The corresponding incidence rates were 3.0 (95% CI 0.6 to 9.7) and 1.6 (95% CI 0.8 to 2.6) per 100 patient years (RR 1.9; 95% CI 0.4 to 6.8; *p* = 0.3). Conversely, 3 (4.7%) of 64 patients with cancer, as compared with 22 (4.1%) of 540 without cancer, had a non-major clinically relevant bleeding event (OR 1.2, 95% CI 0.3 to 3.9, *p* = 0.7). The corresponding incidence rates were 3.0 (95% CI 0.6 to 9.7) and 2.5 (95% CI 1.5 to 3.7) per 100 patient years (RR 1.2; 95% CI 0.2 to 4.0; *p* = 0.7). The respective figures for fatal haemorrhage were 1 (1.6%) vs. 0 (0%).

Overall, 8 (12.5%) of 64 patients with CAT as compared with 39 (7.2%) of 540 without CAT, reported at least one hemorrhagic event during the observation period (OR 1.8, 95% CI 0.8 to 4.1, *p* = 0.1). The corresponding incidence rates were 8.0 (95% CI 3.4 to 15.7) and 4.4 (95% CI 3.1 to 6.0) per 100 patient years (RR 1.8; 95% CI 0.7 to 3.9; *p* = 0.118) (Figure 3).

At the 6-months time point, the standard anticoagulation course for most patients, 5.3% (0.0 to 11.2%) of patients with CAT and 1.3% (0.3 to 2.3%) without CAT had experienced a major bleeding event. The respective figures for non-major clinically relevant bleeding events were 4.1 (0.0 to 9.8) and 2.0 (0.8 to 3.2).

All bleeding events occurred while patients were taking rivaroxaban, irrespective of the presence of cancer at baseline. Most bleedings recorded in CAT patients were at the genitourinary, gastrointestinal, or intracranial site (Table 2).

During the observation period, 1 (1.6%) of 64 patients with CAT died (of fatal bleeding), as compared with 2 (0.4%) of 540 without CAT.

### 4.3. Additional Observations

Anecdotally, due to the limited number of patients available for the multivariate analysis, we observed an association between radiotherapy and recurrent VTE, as well as between chemotherapy and haemorrhage.

To date, we did not observe arterial thromboembolic events.

Appendix A details adverse events occurring in patients with CAT while on anticoagulants.

## 5. Discussion

After a mean follow-up period of one and a half years, we recorded a comparable frequency of recurrent VTE in patients with and without CAT. Interestingly, more than 80% of recurrent events occurred while patients were on rivaroxaban, a proportion which is at all similar to that observed in a subgroup analysis of two large, randomised trials testing rivaroxaban against LMWH plus VKA in CAT patients [17]. This finding both supports and reflects the guideline-recommended strategy of prolonging treatment (so-called indefinite or extended-phase therapy) in patients with persistent risk factors perceived to be at a higher thromboembolic than bleeding risk [18].

Conversely, the proportion of major bleeding and that of all-bleedings was increased in patients with CAT, though not statistically significantly. Of note, all-bleeding events occurred while patients were taking rivaroxaban. This observation matches with data from a classic follow-up study with VKA, in which patients with cancer had a more than doubled 12-month incidence of major bleeding during anticoagulant treatment than patients free of malignancy [1]. Also, the higher frequency of bleedings at the gastrointestinal or genitourinary site appears to be in line with the literature [19,20].

A small number of CAT patients, (4/64, 6%) were treated with prophylactic doses of rivaroxaban (10 mg/day). None of these patients experienced adverse events or died during the study period. Of note, this strategy is in line with the recently published Einstein Choice trial, in which 27/1127 (2.4%) patients randomized to rivaroxaban 10 mg/daily had active cancer [21].

The main limitations of the study are the small sample of cancer patients, the lack of central event adjudication, and the non-randomised design. Nonetheless, we believe our results are robust, because we used a strong methodology, and performed frequent remote monitoring of the study centres and of the data quality through our electronic data capture system [16]. Of note, the 6-month outcome-rate (recurrent VTE, major bleeding, and non-major clinically relevant bleeding) of patients with CAT recorded in the present study overlap with that reported in a recent multicentre, randomised study (SELECT-D) comparing rivaroxaban against dalteparin in patients with CAT, and with that observed in a subgroup analysis of two large randomised trials (EINSTEIN-DVT and EINSTEIN-PE) comparing rivaroxaban against enoxaparin plus VKA to the same purpose (Table 3) [9,19]. Similarly, the 6-month outcome-rate of patients without CAT observed in the present study is similar to that recorded in a pooled analysis, and in a subgroup analysis of two large, randomised trials comparing rivaroxaban against enoxaparin plus VKA for treatment of VTE (EINSTEIN-DVT and EINSTEIN-PE) (Table 3) [17,22].

In conclusion, these preliminary results from the MAC Project suggest that a standard regimen of rivaroxaban is both effective and safe as extended-phase anticoagulation in patients with CAT. Significantly, patients with CAT included in the MAC Project were treated with rivaroxaban for over 18 months on average; that is, three times as much as the mean observation period of patients enrolled in the SELECT-D and EINSTEIN trials.

## Figures and Tables

**Figure 1 life-12-01725-f001:**
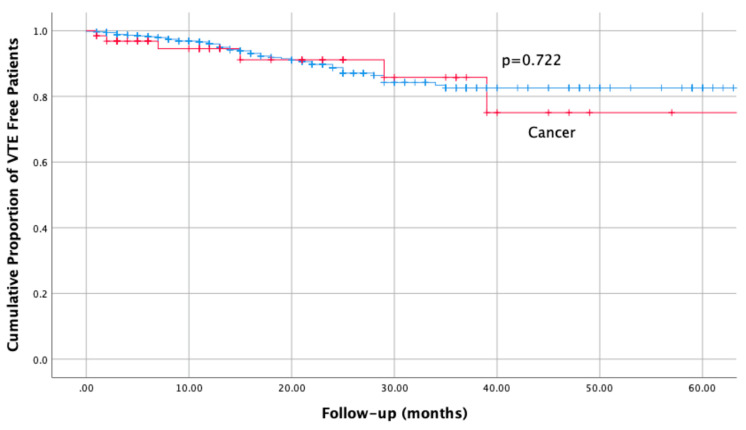
Cumulative incidence of recurrent VTE, in patients with and without cancer at baseline.

**Figure 2 life-12-01725-f002:**
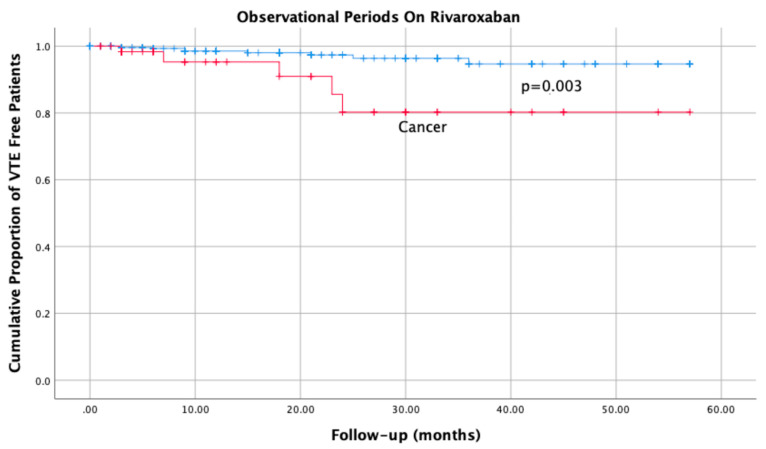
Cumulative incidence of recurrent VTE occurring while on treatment with rivaroxaban, in patients with and without cancer at baseline.

**Figure 3 life-12-01725-f003:**
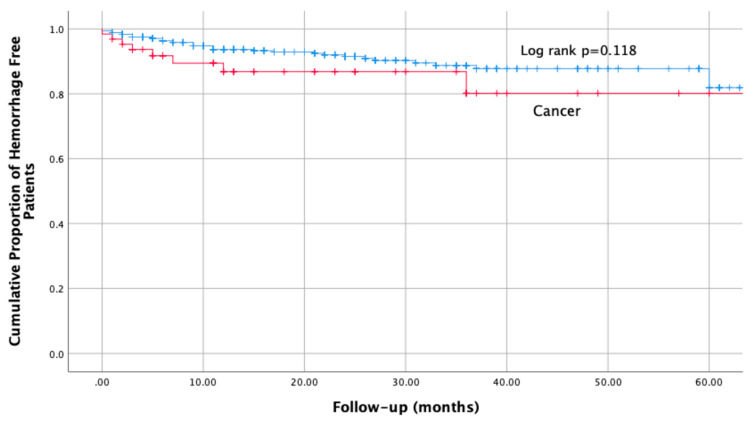
Cumulative incidence of haemorrhage, in patients with and without cancer at baseline.

**Table 1 life-12-01725-t001:** Demographics and metrics.

	Cancer Status		
Variables at Baseline	Absent, *n* = 540	Present, *n* = 64	*p*
Age, years	64.7 (16.4)	71.3 (12.0)	0.002
Weight, kg	74.9 (13.7)	69.3 (14.8)	0.002
Sex (female)	48.3%	56.3%	NS
Previous venous thromboembolism	39.6%	31.3%	NS
Previous major haemorrhage ^A^	2%	1.6%	NS
Heart failure	1.7%	0.0%	NS
COPD	2.2%	3.1%	NS
Diabetes	8.2%	9.4%	NS
Current smoker	15.4%	4.7%	0.022
Hormonal treatment ^B^	3.5%	4.7%	NS
Antiplatelets / NSAIDS before inclusion	12.3%	12.7%	NS
Anticoagulation before inclusion ^C^	35.5%	46.9%	0.099
Follow-up, months	19.8 (16.2)	18.8 (16.1)	NS
Creatinine clearance, mL/min	79.1 (33.3)	61.6 (25.1)	0.001
Haemoglobin levels, g/L	136.9 (18.1)	127.7 (15.9)	0.001
Hematocrit, %	41.4 (4.7)	38.4 (5.2)	0.001
Platelets, ×10^9^/L	236.9 (73.9)	244.8 (92.2)	NS
Alanine Aminotransferase, U/L	23.9 (18.9)	23.1 (16.6)	NS
International normalised ratio	1.2 (0.4)	1.1 (0.4)	NS
APTT, seconds	29.5 (11.8)	24.9 (6.3)	0.045

All data expressed as mean (standard deviation), except otherwise reported. NS: not significant; COPD: chronic obstructive pulmonary disease; APTT: activated partial thromboplastin time. ^A^ including intracranial haemorrhage; ^B^ including hormone replacement therapy or contraception; ^C^ including VKA, LMWH and NOACS.

**Table 2 life-12-01725-t002:** Site of bleeding in patients with or without active cancer at baseline.

	Active Cancer at Baseline	
Site of Bleeding, *n*. (%)	Yes, *n* = 64	No, *n* = 540
Intracranial	2 (3.1)	2 (0.4)
Intraocular	0	1 (0.2)
Upper gastrointestinal	2 (3.1)	5 (0.9)
Lower gastrointestinal	1 (1.6)	7 (1.3)
Vaginal	0	11 (2.0)
Urinary	4 (6.2)	2 (0.4)
Epistaxis	0	3 (0.6)
Gingival	0	2 (0.4)
Muscular	1 (1.6)	5 (0.9)
Other	2 (3.1)	2 (0.4)
Total ^A^	12 (18.7)	40 (7.4)

^A^ There were 4 patients who experienced more than one bleeding event in the CAT group, and 1 in the control group.

**Table 3 life-12-01725-t003:** Comparative frequency of safety and efficacy outcomes in the Mac Project, in the Select-D study, and in the Einstein trials.

Active Cancer			
6-month outcome ^A^	MAC study	Select-D [9]	Einstein pooled, cancer patients [17]
Recurrent VTE	5.5 (0.0 to 11.6)	3.9 (1.7 to 7.6)	4.5 (2.6 to 7.2)
Major bleeding	5.3 (0.0 to 11.2)	2.9 (1.1 to 6.3)	2.3 (0.9 to 4.4)
NMCRB ^B^	4.1 (0.0 to 9.8)	3.5 (1.4 to 6.9)	13.6 (10.2 to 17.6)
**No Cancer**			
6-month outcome ^A^	MAC study	Einstein pooled, main study [22]	Einstein pooled, cancer patients [17] ^C^
Recurrent VTE	1.8 (0.6 to 3.0)	2.1 (1.7 to 2.6)	1.8 (1.4 to 2.3)
Major bleeding	1.3 (0.3 to 2.3)	2.3 (1.8 to 2.8)	1.5 (0.1 to 1.9)
NMCRB ^B^	2.0 (0.8 to 3.2)	7.2 (6.4 to 8.0)	8.9 (7.9 to 9.8)

^A^ All data expressed as % (95% CI). ^B^ Non major clinically relevant bleeding. ^C^ Control group.

## Data Availability

All data supporting reported results can be requested at https://quovadis-ass.it/contact/ accessed 27 October 2022.

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
