# Peer review of "Safety and Efficacy of Rivaroxaban as Extended-Phase Anticoagulation in Patients with Cancer and Venous Thromboembolism: A Preliminary Data Analysis from the Mac Project"

_life, 2022, doi:10.3390/life12111725_

Round 1

Reviewer 1 Report

Dear Authors, 

The following are my comments.

The work is well written, the topic is interesting, and the study well conducted. 

My only great perplexity is the sample too small. 

In addition: 

-       I would remove from the analysis patients taking 10 mg of Rivaroxaban since it is not the correct therapeutic strategy in patients with cancer-related VTE. 

-       What type of cancer was associated with recurrence with or without treatment? What stage? What about prior VTE event in these patients with recurrence?

-       I also suggest reporting the median observation period. 

-       What about arterial thromboembolic event?

-       I would focus specifically on the adverse events that occurred in oncological subjects during anticoagulant therapy by illustrating different clinical parameters in order to provide the reader with more details (e.g. compliance with therapy? Type of chemotherapy? Recent surgery? Hemorrhagic risk factors? Worsening of the disease?). I suggest creating a table with all the information about individual patients and adverse event

I encourage the authors to extend the observation by excluding subjects who take a dosage of Rivaroxaban other than therapeutic in order to obtain more consistent data

Reviewer 2 Report

Thank you for allowing me to review the manuscript entitled:

Safety and Efficacy of Rivaroxaban as Extended-phase Anticoagulation in Patients with Cancer and Venous Thromboembolism: A preliminary data analysis from the Mac Project

I very much enjoyed reading this brief report. It is well known that cancer-associated VTE is a life-threatening and prevalently existing complication in cancer patients. CAT is a complicated situation as prevention and treatment as well have a high risk of adverse events, such as recurrent VTE and major bleeding. It is beneficial to be sure that extended-phase anticoagulation is not harmful but it is protective  and also convenient for the patients.  

Originality/Novelty: It is a brief report for a certain part of the MAC Project that elucidates

extended anticoagulation with NOACs. The duration of follow-up was 18 months and this period is a lot longer than other studies' follow-up for extended anticoagulation

Significance:: It is based on the results of a significant project and even though the number of cancer patients was small the massage of this brief report is important

Quality of Presentation: The presentation is detailed and well organized

Interest to the Readers: It is an important brief report that tries to elucidate the extended anticoagulation in CAT patients

Overall Merit: This brief report supports the idea of extended anticoagulation in CAT patients with a NOAC

 English Level: The English language is accurate

Major Comments

None

Minor Comments

There is no list of the abbreviations used in the text.

Overall, the manuscript has the potential in dealing with a very modern and interesting topic. In my opinion, the article is worthy of publication in Life Journal after minor improvements

Reviewer 3 Report

The paper presents interesting data from routine practice on safety and efficacy of rivaroxaban in cancer patients with VTE. The total number of cancer patients included in the analysis rather small, nevertheless the data is of some importance as any real-life data.

But I have serious concerns regarding data representation.

Major remarks

1.      As I understand some patients, both with and without CAT, did not receive rivaroxaban at the time of recurrent episode. You did not specify how many patients were beyond anticoagulation at this time and for how long. This is more or less probable that in no-CAT group there were more those patients who stopped anticoagulation already. So, I see no reason to compare outcomes in such a different groups. 9.3% vs 8.1% recurrence rate and other similar rates presented in the paper can lead readers to impression that rivaroxaban as effective and safe in CAT patients as in no-CAT patients. But, it may not be true if the majority of no-CAT patients did not receive rivaroxaban at the time of recurrence. It seems you compared those who received rivaroxaban with those who did not.

2.      Moreover, if the goal of the paper is to show efficacy and safety of rivaroxaban then I see no practical point to present data on those w/o CAT.
So, I suggest to present in the paper data only on CAT patients. Another way is to compare thise with Cat on rivaroxaban with those w/o CAT on rivaroxaban.

3.      The Discussion is too short and insufficient. You compared data from MAC that was not presented in Results. In Discussion only data that you presented before has to be discussed. Moreover, you compared your data which is from real-life practice with data from RCTs. But, there are a lot of data published regarding rivaroxaban and other anti-Xa agents in CAT patients in real-life setting. You must discuss your findings comparing with these data also.

Minor remarks

Data on Figure 1 are not that are presented in the text. Compare Lines 115-118 and Figure 1 and you see nothing common. The same is for Figure 2 and the text also.

Lines 89, 91. There is no need to refer to any sources regarding statistical tools. So, references 17 and 18 has to be removed from the list.

Lines 104, 105. It was nor specified before that you were going to use quartiles when presenting data. Please, specify this in Statistical analysis paragraph.

Round 2

Reviewer 1 Report

I am pleased to see that my comments have been welcomed. I would like to revise the manuscript with the changes made in the text as some modifications are missing,

Futhermore:

1) I would remove from the analysis patients taking 10 mg of Rivaroxaban since it is not the correct therapeutic strategy in patients with cancer-related VTE. 

Actually, only 4/64 (6%) patients with cancer-related VTE were on 10 mg.
We believe these patients should not be removed from the analysis, since the MAC Project is not an interventional randomised trial. The aim of the study is to gather information on the long-term outcome of unselected anticoagulated patients. Clinical decisions, including treatment modifications are left to the attending physician. These things happen in the everyday clinical work. 

R: If the authors decide not to remove from the analysis patients treated with 10 mg of Rivaroxaban then they have to emphasize in “discussion” that it is not the correct dosage for the treatment of cancer-related thrombosis and illustrate when such use can be considered also by providing the appropriate published evidence. If there is anything that can be improved in “everyday clinical work”, it should be emphasized in a paper of such scientific value.

2) What type of cancer was associated with recurrence with or without treatment? What stage? What about prior VTE events in these patients with recurrence? 

We tried to resume the data you’ve asked for in the following tables, hopefully you’ll find it satisfying. Unfortunately we haven’t got information on cancer stage in the DB at the moment. 

R: I am satisfied with, thanks for the modification.

3) I also suggest reporting the median observation period. 

The median observation period for patients without cancer-related VTE is 14 months, and for patients without cancer-related VTE is 12.5 months. 

R: Good, please report in the text.

4)  What about arterial thromboembolic events? 

We currently observed no arterial thromboembolic events. 

R: Good, please report in the text.

5) I would focus specifically on the adverse events that occurred in oncological subjects during anticoagulant therapy by illustrating different clinical parameters in order to provide the reader with more details (e.g. compliance with therapy? Type of chemotherapy? Recent surgery? Hemorrhagic risk factors? Worsening of the disease?). I suggest creating a table with all the information about individual patients and adverse event 

Dear reviewer, we of course can create such a table; however, honestly, it’ll really take a lot of work. If you really think this is mandatory for the manuscript to improve, we’ll do it, but we need a little more time. 

R: Adverse events (bleeding, thrombosis) that occur during treatment with DOAC, especially in cancer patients, are often due to a trigger that is beyond the effectiveness/safety of DOAC (i.e. chemotherapy, progression of oncological disease, fasting, thrombocytopenia, etc.). Since the adverse events reported are numerically few I think it is extremely useful to report for single event (hemorrhagic and re-thrombosis) which are the possible causes of the event. Definitely, time well spent.

Reviewer 3 Report

Dear Authors!

Thank you for addressing my remarks and for clarifying your point of view on what triggered my serious concerns.

I can get your point regarding comparison of CAT and non-CAT patients. You intended to show that CAT patients on DOACs had the same outcomes rates as non-CAT patients not on DOACs, that’ ok. But, please either present this as one of the study aims or in the Material and Methods section that you were going not just register outcomes, but were also going to compare two sub-groups.

Regarding 6-months results of the MAC project that you discussed in Discussion, please, present them in the Results section also.

Round 3

Reviewer 1 Report

I thank the authors for the changes that I think have improved the manuscript. I have no further comments.